# Risk factors of ectopic pregnancy after in vitro fertilization-embryo transfer in Chinese population: A meta-analysis

Yanbo Wang *, Li Chen, Yuan Tao, Mengqian Luo

School of Nursing, Gansu University of Traditional Chinese Medicine, Lanzhou, Gansu Province, China

* cl1343440599@126.com

## Abstract

### Background

The prevalence of ectopic pregnancy after assisted reproduction is notably high, posing a significant threat to the life safety of pregnant women. Discrepancies in published results and the lack of a comprehensive description of all risk factors have led to ongoing uncertainties concerning ectopic pregnancy after assisted reproduction.

### Objective

This study aimed to understand the risk factors for ectopic pregnancy after in vitro fertilization-embryo transfer in the Chinese population and provide a reference for targeted prevention and treatment.

### Methods

A comprehensive search of the China National Knowledge Infrastructure, Wang fang Database, China Science Technology Journal Database, Chinese Biomedical Literature Database, PubMed, Web of Science, and Embase was conducted to identify relevant literature on the risk factors for ectopic pregnancy in Chinese women after assisted reproductive technology in Chinese women. A meta-analysis of the included studies was performed using Stata17.

### Results

Overall, 34 articles were included in the analysis. The risk factors for ectopic pregnancy after in vitro fertilization-embryo transfer in the Chinese population included a thin endometrium on the day of HCG administration and embryo transplantation, a history of ectopic pregnancy, secondary infertility, a history of induced abortion, polycystic ovary syndrome, decreased ovarian reserve, tubal factor infertility, cleavage stage embryo transfer, fresh embryo transfer, artificial cycle protocols, elevated estradiol levels on the day of human chorionic gonadotropin administration, a history of tubal surgery, two or more number of embryo transfers, previous pregnancy history, and a history of pelvic surgery.

**Data Availability Statement:** All relevant data are within the paper and its Supporting information files.

**Funding:** This study was supported by Education Technology Innovation Project in Gansu Province

(No:2022A-069). The funders had no role in study design, data collection and analysis, decision to publish,or preparation of the manuscript.

**Competing interests:** The authors have declared that no competing interests exist.

## Conclusion

This study clarified the factors influencing ectopic pregnancy after in vitro fertilization and embryo transfer in the Chinese population, focusing on high-risk groups. Targeted and personalized intervention measures should be adopted to prevent and detect the disease early to reduce its incidence and harm.

## Trial registration

The protocol for this view was registered in PROSPERO (CRD42023414710).

## Introduction

In recent years, there has been a notable increase in the annual incidence of infertility. According to statistics, one out of every seven couples of childbearing age experience infertility [1]. The rapid development and widespread application of in vitro fertilization-embryo transfer (IVF-ET) have become important methods for infertile couples to achieve fertility. However, ectopic pregnancy (EP), a high-risk complication of IVF-ET, occasionally occurs. EP is not only a pregnancy failure but also a direct threat to the patient's life. The incidence of EP after IVF-ET in China is between 3.2% and 8.6% [2, 3], which is significantly higher than that observed after natural conception [4]. There are many related original studies; however, their results differ [5, 6].

Furthermore, only a few studies have used meta-analyses to quantitatively and systematically evaluate these findings. Therefore, this study aimed to analyze the risk factors for EP after IVF-ET in the Chinese population using evidence-based medicine and provide a reference for identifying high-risk groups and implementing targeted prevention. We have successfully achieved this aim.

## Materials and methods

### Search strategy

The meat-analysis, was performed following the Preferred Reporting Items for Systematic Reviews and Meta-Analyses (PRISMA) guidelines; the PRISMA Checklist is presented in S1 Checklist. We systematically searched the China National Knowledge Infrastructure, Wang fang Database, China Science Technology Journal Database, Chinese Biomedical Literature Database, PubMed, Web of Science and Embase databases from their establishment to April 2023. We sought relevant literature on the risk factors associated with EP after IVF-ET in Chinese women. Additionally, the references within the selected studies were reviewed. A combination of subject-specific terms and free-text keywords was used for retrieval and adjusted according to different databases. The search terms included "assisted reproductive technology", "in vitro fertilization and embryo transfer", "ectopic pregnancy", "risk factors", "influencing factors", "related factors".

### Inclusion/Exclusion criteria

Studies were included in the meta-analysis if: (1) the study participants were Chinese women who had undergone IVF-ET; (2) the fertilization technique used was IVF or intracytoplasmic sperm injection (ICSI); (3) the original literature used multivariate logistic regression analysis to identify relevant risk factors; (4) the study type was case-control or cohort study; (5) they

   

were published in Chinese or English; (6) literature quality score was $\geq 7$ points; (7) there were clear definitions of cases and risk factors in the literature; (8) the number of studies on the same risk factor was $\geq 2$; and (9) if the same study population was reported in different articles, articles with more risk factors were included. The exclusion criteria employed were as follows: (1) literature aimed at studying the risk factors for heterotopic pregnancy or recurrent EP; (2) literature with the same data published repeatedly or in different articles; (3) inaccessibility to full text, invalid data, data self-contradictory literature; (4) studies involving unconventional techniques, such as preimplantation embryo genetic diagnosis and intratubal transplantation.

### Study selection, quality evaluation, and data extraction

The retrieved literature was imported into Endnote, a literature management software. After removing duplicate studies, the remaining literature was screened according to the inclusion and exclusion criteria to determine whether they were included. The Newcastle-Ottawa Scale was used to evaluate the quality of the literature that might be included. Finally, the included literature was determined, and data were extracted, including the author's name, year of publication, country of publication, study type, sample size, and identified risk factors. Two researchers independently executed the entire process after unified training, and the results were cross-checked. The part with differences in the results was decided by the third person.

### Statistical analysis

We used Stata17 for statistical analysis. Odds ratio (OR) and their 95% confidence interval (CI) were used to represent the effects of the statistical analysis. Heterogeneity between the studies was evaluated using $I^2$ values. An $I^2$ value of $\leq 50\%$ indicated low heterogeneity, prompting the selection of a fixed effect model for statistical aggregation. Otherwise, indicated high heterogeneity, prompting the selection of a random effect model for statistical aggregation. The sensitivities of the results were evaluated by transforming the two analysis models. The Egger's test was used to analyze publication bias for risk factors with more than nine articles. The stability of the results for risk factors with publication bias was evaluated using the clipping method. Statistical significance was set at $P < 0.05$.

## Results

### The selection of study

Initially, a total of 1,786 articles were retrieved, and after screening based on the inclusion and exclusion criteria, 34 articles were finally included in the study. The selection process is shown in S1 Fig. All data included in the literature can be seen in S1 Data.

### Basic characteristics and quality evaluation of the included literature

Among the 34 articles, 25 were case-control studies, and nine were cohort studies. The sample sources involved 16 provinces and municipalities across the country. In this study, the research data from Zhu BY [7], Wang [8], and Zhang CC [9] were divided into fresh cycle and freeze-thaw cycle categories, resulting in two separate data extractions. The basic characteristics of the included studies are presented in Table 1.

### Results of meta-analysis

Heterogeneity tests showed that infertility type, history of induced abortion, polycystic ovarian syndrome (PCOS), decreased ovarian reserve, thawed endometrial preparation plan, estradiol

**Table 1. Basic characteristics of included literature.**

| Senior author and Publication year | Study site | Research Type | EP | NOT EP | Risk Factor | NOS Score |
|---|---|---|---|---|---|---|
| Zeng QL [10] 2022 | Hunan | A | 170 | 1999 | 1, 2, 6, 12, 18 | 7 |
| Guo XH [11] 2018 | Gansu | A | 170 | 170 | 2, 9, 10, 11, 12, 19, 23 | 7 |
| Hu WH [12] 2018 | Sichuan | A | 50 | 50 | 1, 3, 14 | 7 |
| Kong HJ [1] 2021 | Henan | A | 214 | 12552 | 4, 9, 10, 16 | 8 |
| Li CM [13] 2020 | Hubei | A | 100 | 100 | 1, 5, 9, 14 | 7 |
| Li L [14] 2018 | Guangdong | A | 196 | 8352 | 2, 7, 9, 10, 11 | 7 |
| Li Q [15] 2020 | Guangdong | A | 20 | 20 | 1, 2, 14 | 7 |
| Pan R [16] 2018 | Shanxi | A | 76 | 5489 | 1, 9, 16, 17, 19 | 7 |
| Wang H [17] 2016 | Shandong | A | 74 | 2038 | 2, 4, 23 | 7 |
| Wang YN [18] 2020 | Jiangsu | A | 64 | 64 | 2, 7, 10, 11 | 7 |
| Yang YJ [19] 2015 | Shanghai | A | 97 | 97 | 1, 3, 14 | 7 |
| Zhang CC [9]a 2019 | Qinghai | A | 25 | 425 | 13, 23, 24 | 7 |
| Zhang CC [9]b 2019 | Qinghai | A | 22 | 418 | 16, 23 | 7 |
| Zhang H [20] 2012 | Guangxi | A | 35 | 361 | 2, 9, 24 | 7 |
| Zhang Y [21] 2018 | Shanghai | A | 80 | 80 | 1, 3, 14 | 7 |
| Zhu BY [7]a 2021 | Hebei | A | 18 | 942 | 2, 4, 13, 23 | 8 |
| Zhu BY [7]b 2021 | Hebei | A | 30 | 1960 | 2, 4, 16, 23 | 8 |
| Han JC [22] 2021 | Tianjin | A | 58 | 58 | 1, 5, 9 | 7 |
| Zheng JH [23] 2019 | Hebei | A | 7 | 28 | 14 | 8 |
| Jiang HM [24] 2017 | Hubei | A | 148 | 148 | 3, 14 | 7 |
| Zhou Y [25] 2014 | Zhejiang | A | 39 | 1525 | 3, 11, 15 | 7 |
| Weng D [2] 2023 | Shanxi | A | 45 | 477 | 2, 13 | 8 |
| Bu [3] 2016 | Henan | B | 538 | 16139 | 9,10, 11, 16 | 8 |
| Fang [26] 2021 | Guangdong | A | 92 | 3025 | 1, 9, 15, 17 | 8 |
| Huang [27] 2020 | Shanghai | B | 18 | 847 | 18 | 9 |
| Jin [28] 2020 | Zhejiang | A | 278 | 13142 | 1, 2, 9, 10 | 8 |
| Jing [29] 2019 | Hunan | B | 119 | 10244 | 12 | 8 |
| Lin [30] 2017 | Beijing | B | 93 | 2253 | 2, 3, 5, 8, 15, 16, 17, 20, 21, 22, 24 | 8 |
| Liu [31] 2020 | Shanghai | B | 543 | 16701 | 1, 7, 9, 10, 12, 22 | 8 |
| Liu [5] 2019 | Shanxi | A | 225 | 900 | 1, 2, 3, 9, 10, 11, 16, 17, 18, 19, 20, 21, 22, 23, 24 | 7 |
| Zhang [6] 2017 | Beijing | B | 45960 | 23796 | 1, 3, 8, 9, 10, 11, 15, 21 | 7 |
| Zhao [32] 2022 | Hunan | A | 183 | 5777 | 1, 2, 9, 10, 21 | 7 |
| Wang [8]a 2013 | Jiangsu | B | 83 | 5256 | 6, 9 | 8 |
| Wang [8]b 2013 | Jiangsu | B | 40 | 1996 | 6 | 8 |
| Wen [4] 2022 | Jiangsu | A | 67 | 2984 | 10, 16 | 9 |
| Liu [33] 2022 | Henan | B | 214 | 12552 | 4, 9, 10, 16 | 9 |
| Hu [34] 2022 | Shanghai | B | 336 | 15665 | 11, 19 | 9 |

a = Fresh, b = Frozen; 1 = EMT on HCG administration day, 2 = History of EP, 3 = Infertility type, 4 = EMT at transplantation, 5 = History of induced abortion, 6 = PCOS, polycystic ovary syndrome, 7 = Male factor infertility, 8 = DOR, diminished ovarian reserve, 9 = Tubal factor infertility, 10 = Embryo transfer stage, 11 = Type of transfer, 12 = Endometrial preparation, 13 = E2 level on HCG day, 14 = Previous tubal surgery, 15 = Maternal age, 16 = No. of transferred embryos, 17 = Fertilization method, 18 = Previous of cesarean section, 19 = Ovulation Protocol, 20 = No. of oocytes retrieved, 21 = Maternal BMI, 22 = Previous pregnancy, 23 = History of pelvic surgery, 24 = Dose of gonadotrophin (IU).

(E2) level on the day of HCG administration, history of tubal surgery, history of cesarean section, number of oocytes aspirated, maternal body mass index (BMI), previous pregnancy history, and total dose of gonadotropin were less heterogeneous among the literature; thus, a fixed effect model was used for consolidation. A heterogeneity test of other factors included in

the literature showed that $I^2 > 50\%$, indicating significant heterogeneity. Therefore, a random effects model was used for consolidation. The results of the meta-analysis showed that a thin endometrium on the days of HCG administration and embryo transfer (OR 1.951, 95% CI [1.598–2.381]), (OR 1.511, 95% CI [1.197–1.908]), history of EP (OR 1.541, 95%CI [1.213–1.957]), secondary infertility (OR 1.326, 95% CI [1.171–1.502]), history of induced abortion (OR 2.054, 95% CI [1.310–3.222]), PCOS (OR 2.164, 95% CI [1.386–3.381]), decreased ovarian reserve (OR 1.751, 95% CI [1.346–2.279]), tubal factor infertility (OR 1.851, 95% CI [1.609–2.130]), cleavage stage embryo transfer (OR 1.870, 95%CI [1.417–2.466]), fresh embryo transfer (OR 1.463, 95% CI [1.062–2.016]), artificial cycle (OR 2.067, 95% CI [1.718–2.487]), higher E2 level on HCG day (OR 1.001, 95% CI [1.001–1.001]), history of fallopian tube surgery (OR 2.692, 95% CI [2.075–3.494]), two or more number of embryo transfer (OR 1.517, 95% CI [1.226–1.878]), history of cesarean section (OR 1.632, 95% CI [1.005–2.652]), past pregnancy history (OR 1.227, 95% CI [1.057–1.423]) and history of pelvic surgery (OR 1.909, 95% CI [1.349–2.701]) were identified as risk factors for EP after IVF-ET. Additionally, maternal age and male factor infertility were found to be associated with EPs. The fertilization method, ovulation induction protocol, number of aspirated oocytes, maternal BMI, and total dose of gonadotropin were unrelated to EP after IVF-ET. The detailed results of the meta-analysis are shown in Table 2. A forest plot example based on the type of infertility is shown in S2 Fig.

**Table 2. Heterogeneity test and meta analysis results.**

| Risk Factor | | reference group | Number of studies | Heterogeneity | | Pooling Model | Meta-analysis results | |
|---|---|---|---|---|---|---|---|---|
| | | | | P-value | I2 (%) | | OR (95%CI) | P-value |
| EMT on HCG day | Thin | Thick | 14 | <0.001 | 71.2 | Random | 1.951 (1.598–2.381) | <0.001 |
| History of EP | Yes | No | 14 | <0.001 | 79.3 | Random | 1.541 (1.213–1.957) | <0.001 |
| Infertility type | Secondary | Primary | 8 | 0.083 | 44.4 | Fixed | 1.326 (1.171–1.502) | <0.001 |
| EMT at transplantation | Thin | Thick | 5 | 0.049 | 58.1 | Random | 1.511 (1.197–1.908) | 0.001 |
| Previous miscarriage | Yes | No | 3 | 0.937 | 0 | Fixed | 2.054 (1.310–3.222) | 0.002 |
| PCOS | Yes | No | 3 | 0.377 | 0 | Fixed | 2.164 (1.386–3.381) | 0.001 |
| Male factor infertility | Yes | No | 3 | 0.078 | 60.9 | Random | 0.534 (0.342–0.834) | 0.006 |
| DOR | Yes | No | 2 | 0.359 | 0 | Fixed | 1.751 (1.346–2.279) | <0.001 |
| Tubal factor infertility | Yes | No | 17 | 0.001 | 59.3 | Random | 1.851 (1.609–2.130) | <0.001 |
| Embryo transfer stage | Cleavage | Blastocyst | 12 | <0.001 | 83.1 | Random | 1.870 (1.417–2.466) | <0.001 |
| Type of transfer | Fresh | Frozen | 8 | <0.001 | 88.5 | Random | 1.463 (1.062–2.016) | 0.020 |
| Endometrial preparation | Artificial | Natural | 4 | 0.756 | 0 | Fixed | 2.067 (1.718–2.487) | <0.001 |
| E2 level on HCG day | High | Low | 3 | 0.192 | 39.4 | Fixed | 1.001 (1.001–1.001) | <0.001 |
| Previous tubal surgery | Yes | No | 7 | 0.478 | 0 | Fixed | 2.692 (2.075–3.494) | <0.001 |
| Maternal age | ≥30 | <30 | 4 | 0.002 | 79.7 | Random | 0.935 (0.880–0.993) | 0.029 |
| No. of transferred embryos | ≥2 | <2 | 10 | 0.003 | 63.8 | Random | 1.517 (1.226–1.878) | <0.001 |
| Fertilization method | IVF | ICSI | 4 | 0.043 | 63.2 | Random | 1.122 (0.734–1.715) | 0.595 |
| Previous of cesarean section | Yes | No | 3 | 0.338 | 7.8 | Fixed | 1.632 (1.005–2.652) | 0.048 |
| Ovulation Protocol | Antagonist | Agonist | 4 | 0.021 | 69.1 | Random | 1.193 (0.677–2.102) | 0.542 |
| No. of oocytes retrieved | - | - | 2 | 0.385 | 0 | Fixed | 1.020 (0.994–1.047) | 0.138 |
| Maternal BMI | - | - | 4 | 0.765 | 0 | Fixed | 1.001 (1.000–1.002) | 0.056 |
| Previous pregnancy | Yes | No | 3 | 0.297 | 17.7 | Fixed | 1.227 (1.057–1.423) | 0.007 |
| History of pelvic surgery | Yes | No | 7 | 0.005 | 67.5 | Random | 1.909 (1.349–2.701) | <0.001 |
| Dose of gonadotrophin (IU) | - | - | 4 | 0.657 | 0 | Fixed | 1.530 (0.940–2.490) | 0.087 |

EMT:endometrial thickness; HCG:human chorionic gonadotropin; EP:ectopic pregnancy; PCOS:polycystic ovarian syndrome; DOR:diminished ovarian reserve; E2: estradiol; BMI:body mass index.

## Sensitivity analysis

Statistically significant influencing factors were analyzed using both random and fixed effects models. The results showed that a history of cesarean section was a significant risk factor for EP in the fixed effects model. However, under the random-effects model, with P = 0.062, it was no longer statistically significant, suggesting that the results were unstable; further investigation is needed to determine whether a history of cesarean section is indeed a risk factor for EP after IVF-ET. The other risk factors did not change significantly between the two effect models, and the combined results were stable. The specific values are listed in Table 3.

## Publication bias

More than nine articles included endometrial thickness on HCG administration day, history of EP, tubal factor infertility, cleavage stage embryo transfer, and the number of embryos transferred. Publication bias was assessed using Egger's test. The results indicated the possibility of publication bias concerning endometrial thickness on the day of HCG administration. In contrast, the possibility of publication bias for other risk factors was small, the specific values are shown in Table 3. Publication bias regarding endometrial thickness on the day of HCG administration was corrected using the clipping method. The corrected results showed that after the data from eight virtual studies were included, the combined result was OR = 1.390, with a 95% CI of 1.156–1.672, which was not significantly different from the value before correction, indicating that this result was minimally affected by publication bias.

**Table 3. Sensitivity analysis and publication bias results.**

| Risk Factor | Fixed pooling model | | Random pooling model | | egger's test P-value |
|---|---|---|---|---|---|
| | OR (95%CI) | P-value | OR (95%) | P-value | |
| EMT on HCG day | 1.337 (1.281–1.395) | <0.001 | 1.951 (1.598–2.381) | <0.001 | <0.001 |
| History of EP | 1.291 (1.184–1.409) | <0.001 | 1.541 (1.213–1.957) | <0.001 | 0.127 |
| Infertility type | 1.326 (1.171–1.502) | <0.001 | 1.566 (1.224–2.004) | <0.001 | - |
| EMT at transplantation | 1.345 (1.189–1.521) | <0.001 | 1.511 (1.197–1.908) | 0.001 | |
| Previous miscarriage | 2.054 (1.310–3.222) | 0.002 | 2.054 (1.310–3.222) | 0.002 | |
| PCOS | 2.164 (1.386–3.381) | 0.001 | 2.164 (1.386–3.381) | 0.001 | |
| Male factor infertility | 0.508 (0.387–0.668) | <0.001 | 0.534 (0.342–0.834) | 0.006 | |
| DOR | 1.751 (1.346–2.279) | <0.001 | 1.751 (1.346–2.279) | <0.001 | |
| Tubal factor infertility | 1.724 (1.596–1.862) | <0.001 | 1.851 (1.609–2.130) | <0.001 | 0.060 |
| Embryo transfer stage | 1.892 (1.695–2.112) | <0.001 | 1.870 (1.417–2.466) | <0.001 | 0.843 |
| Type of transfer | 1.422 (1.283–1.576) | <0.001 | 1.463 (1.062–2.016) | 0.02 | |
| Endometrial preparation | 2.067 (1.718–2.487) | <0.001 | 2.067 (1.718–2.487) | <0.001 | |
| E2 level on HCG day | 1.001 (1.001–1.001) | <0.001 | 1.001 (1.000–1.002) | <0.001 | |
| Previous tubal surgery | 2.692 (2.075–3.494) | <0.001 | 2.692 (2.075–3.494) | <0.001 | |
| Maternal age | 0.973 (0.956–0.990) | 0.002 | 0.935 (0.880–0.993) | 0.029 | |
| No. of transferred embryos | 1.364 (1.219–1.526) | <0.001 | 1.517 (1.226–1.878) | <0.001 | 0.063 |
| Previous of cesarean section | 1.632 (1.005–2.652) | 0.048 | 1.621 (0.976–2.692) | 0.062 | |
| Previous pregnancy | 1.227 (1.057–1.423) | 0.007 | 1.205 (1.008–1.442) | 0.041 | |
| History of pelvic surgery | 1.447 (1.277–1.640) | <0.001 | 1.909 (1.349–2.701) | <0.001 | |

EMT:endometrial thickness; HCG:human chorionic gonadotropin; EP:ectopic pregnancy; PCOS:polycystic ovarian syndrome; DOR:diminished ovarian reserve; E2: estradiol.

## Discussion

The results of this study showed that a thin endometrium on the days of HCG administration and embryo transfer is a risk factor for EP after IVF-ET, which may be related to the fact that the embryo is implanted closer to the spiral artery with a higher oxygen concentration in the uterine cavity with a thin endometrium. High oxygen concentrations inhibit embryonic development and force the embryo to implant in locations outside the uterus with lower oxygen concentrations [32]. In essence, a thin endometrium has low receptivity, which is not conducive to the normal implantation of embryos; however, the current study could not determine the ideal endometrial thickness for optimal embryo implantation development.

Decreased ovarian reserve and PCOS are also risk factors for EP after IVF-ET. This connection may be related to ovarian endocrine dysfunction and dysregulation of estrogen and progesterone levels in these patients, resulting in poor endometrial receptivity formed by the combined action of estrogen and progesterone [35]. Tubal factor infertility, a common reasons for seeking assisted reproductive pregnancy, also poses a risk factor for EP after assisted pregnancy. This is mainly related to impaired tubal peristalsis and transport capacity due to tubal lesions. In theory, IVF-ET technology directly places embryos in the uterine cavity, and embryos should not be implanted elsewhere. However, this is usually not the case. Some studies [36] have found that part of the culture medium injected into the uterine cavity may flow into other areas, such as the fallopian tubes or cervix. When tubal function is destroyed, embryos entering the fallopian tube with the culture medium are retained and implanted, thus forming an EP.

A history of fallopian tube and pelvic surgery can damage the fallopian tube to varying degrees, cause fallopian tube dysfunction, and increase the risk of EPs. The results of this study suggest that previous pregnancy, whether ectopic or not, is a risk factor for EP after IVF-ET. A possible reason is that the patient's reproductive system has not been cured, there is still inflammation, or the patient's own untreated EP risk factors that interfere with the successful implantation of transplanted embryos. The results of the meta-analysis also suggest that abortion is also a risk factor for EP after IVF-ET, which is consistent with Wang Hu's study [37] that once pregnancy ends in any form, it can cause endometrial damage and inflammation, and increase the risk of EP. Compared with primary infertility, secondary infertility is also a risk factor for EP after IVF-ET and is mainly caused by a history of induced abortion and pelvic tubal disease.

This study indicates that cleavage-stage embryo transfer increases the risk of EP after IVF-ET compared to blastocyst-stage embryo transfer, which is consistent with previous research [38]. Blastocyst-stage embryos have a higher degree of hatching and a shorter time from implantation to the uterus, which reduces the possibility of EP caused by the migration of embryos to other parts, while cleavage-stage embryos are just the opposite; and 7 days after HCG administration, the reverse contraction wave of the uterus from the cervix was significantly weakened or even disappeared, when the blastocyst-stage embryo moved into the uterus, the intensity of the uterine contraction wave is lower than when the cleavage stage was transplanted, while the volume of the blastocyst-stage embryo is more resistant to the contraction of the wave than in the cleavage stage, compared with the cleavage embryo stage is more likely to be squeezed and migrated to other parts forming EP [39]. Contrary to previous studies [40] showing no difference in the incidence of EP between single embryo transfer and multiple embryo transfer, this meta-analysis suggests that two or more embryo implantations increase the risk of EP after IVF-ET. Increasing the number of embryo transfers increases the probability of embryo migration to other parts of the body, consequently increasing the risk of EP.

The study's findings also highlight that fresh cycle transplantation is a risk factor for EP after IVF-ET. High dose of gonadotropin and abnormal hormone environment may be the cause of abnormal embryo implantation during fresh cycle transplantation [41]. In the freeze-thaw cycles, the artificial cycle endometrial preparation plan is a risk factor for EP, mainly affecting patients with irregular ovulation and poor endometrial receptivity [9]. Moreover, a high serum concentration of E2 on the day of HCG administration is also a risk factor for EP. Excessive E2 levels reduce endometrial receptivity and interfere with normal peristalsis of the fallopian tube, impairing embryo implantation and reverse embryo transfer [2, 42].

This study does not definitively determine whether a history of cesarean section is a risk factor for EP after IVF-ET due to the poor stability of the available data. Therefore, future research needs to include more data to confirm this relationship.

The results of this meta-analysis also suggest that male infertility reduces the risk of EP after IVF-ET. Furthermore, maternal age is a protective factor against EP. This may be because younger patients are more sensitive to ovulation induction and have higher levels of E2 in their bodies, which leads to a poor hormonal environment and increases the incidence of EP [43]. It may also be that more young patients underwent IVF-ET, resulting in younger patients in the EP group and a deviation.

Moreover, the literature included in this study [11, 24] showed that the fallopian tubes are the most common site for EP after IVF-ET, similar to natural pregnancy, with over 87% of cases occurring there. Among these, the incidence of EP in the ampulla of the fallopian tubes was the highest, accounting for over 70% of all EPs. The incidence of interstitial pregnancy in the fallopian tube is only 0.8%. However, their occurrence is ten times higher than that in natural conception, with a mortality rate reaching 2–3%, which is seven times higher than that of other ordinary fallopian tube pregnancies [44]. Other rare types of EPs that occur outside the fallopian tubes after IVF-ET include cornual, ovarian, scar, abdominal, and cervical pregnancies. Although the incidence of EPs is less than 0.1%, the mortality rate is high, particularly for abdominal pregnancies, which can reach 20% [45, 46]. There is a significant upward trend in the proportion of EPs in these special areas after IVF-ET [11, 21, 24], and timely diagnosis and treatment are key to saving the lives of pregnant women. Ultrasound combined with $\beta$-HCG measurement is still the main method for diagnosing EP [47].

Conservative treatment has become the first-line treatment for patients with fertility needs after an EP. It can be divided into two methods: medication, such as methotrexate, mifepristone, and misoprostol, and conservative surgical treatment. Methotrexate, in particular, is recognized as a safe and effective drug for treating EP both nationally and internationally [48]. In clinical practice, EP patients with fertility needs and stable conditions, without ruptured gestational sac or abdominal bleeding, may consider conservative treatment with local or intramuscular injection of methotrexate. Still, attention should be paid to monitoring $\beta$- HCG levels. In cesarean scar pregnancy, it is recommended to combine local injection and systemic intramuscular injection of methotrexate for medication [47]. Laparoscopic surgery, a conservative treatment method with minimal trauma, excellent efficacy, and rapid recovery, is recommended in clinical practice. Laparoscopic corneostomy and suturing can preserve the complete morphology of the fallopian tubes and uterus, making it the mainstream conservative surgical treatment for interstitial pregnancy. However, they are prone to residual trophoblastic cells, and regular monitoring of blood HCG levels should be performed after surgery until it normalizes [48]. When a cornual pregnancy occurs, conservative treatment with medication is not recommended if only a small portion of the gestational sac is located within the uterine cavity. In such cases, it is recommended to opt for either hysteroscopic electro-resection or a laparoscopic cornual incision to facilitate embryo retrieval under laparoscopic monitoring [49]. For patients with ovarian pregnancy and unstable hemodynamics, laparoscopy is recommended as

the first choice for conservative treatment. Patients with fertility needs should consider undergoing ovarian wedge resection to preserve as much of the ovarian reserve as possible [46]. Conservative surgical treatments for patients with cesarean scar pregnancy include curettage, hysteroscopy, laparoscopic surgery to remove the lesion, and vaginal scar pregnancy resection [50].

Currently, reliable data on optimal management strategies for abdominal pregnancies are lacking. A previous study reported a case of primary liver pregnancy successfully treated with laparoscopic exploration combined with postoperative intramuscular methotrexate injection [51]. There have also been successful cases of conservative drug treatment using intramuscular injections of methotrexate [52]. However, due to the high risk of abdominal pregnancy and the possibility of rupture of the gestational sac at any time, surgical intervention remains the primary choice. For cervical pregnancy, the optimal treatment approach remains unclear; for patients unresponsive to drug treatments with high bleeding volumes, uterine artery embolization can be considered; Surgical treatment is no longer just a simple hysterectomy, and minimally invasive surgical lesion resection can also be used as a conservative treatment option for cervical pregnancy [46, 47]. Experts also recommend laparoscopic surgery for patients with EPs in the fallopian tubes. Generally, laparoscopic salpingectomy or salpingotomy is performed to remove embryos. Patients with normal fallopian tubes on the other side and reproductive needs can also undergo salpingectomy. If the other side of the fallopian tube is damaged, salpingotomy or salpingectomy can be considered based on the extent of damage to the pregnancy-affected fallopian tube [53].

The findings of this review serve as an evidence-based foundation for understanding the risk factors associated with EP after IVF-ET in the Chinese population and provide a reference for the early prevention of EP after IVF-ET and the identification of high-risk groups. Nevertheless, this study also has some limitations. First, the number of studies that included certain risk factors was small, necessitating further validation through additional research. Second, heterogeneity among the included studies may have affected the effectiveness of the statistical analysis. Finally, the participants included in this study were only Chinese women, which limits the generalizability of the research findings. Despite these limitations, our findings make an important contribution to understanding a wide range of acceptable risks for EP after IVF-ET and provide directions for future research by addressing the shortcomings of this study.

## Conclusion

The risk factors for EP after IVF-ET are a thin endometrium on the days of HCG administration and embryo transfer, history of EP, secondary infertility, history of induced abortion, PCOS, decreased ovarian reserve, tubal factor infertility, cleavage-stage embryo transfer, fresh embryo transfer, artificial cycle, high E2 level on the day of HCG administration, history of tubal surgery, two or more embryo transfers, previous pregnancy history, and pelvic surgery history.

## Supporting information

**S1 Fig. Flow chart of literature screening.**
(TIF)

**S2 Fig. Infertility type forest plot.**
(TIF)

**S1 Checklist. PRISMA checklist.**
(DOCX)

**S1 Data. Include all literature data.**
(XLS)

## Acknowledgments

We would like to thank Editage ([www.editage.cn](www.editage.cn)) for English language editing.

## Author Contributions

**Data curation:** Li Chen, Yuan Tao, Mengqian Luo.

**Funding acquisition:** Yanbo Wang.

**Methodology:** Yanbo Wang.

**Software:** Li Chen, Yuan Tao.

**Writing – original draft:** Li Chen.

**Writing – review & editing:** Yanbo Wang, Li Chen.

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
