## [Decision Letter · Decision Letter 0]

27 Oct 2023

PONE-D-23-29164Risk factors of ectopic pregnancy after IVF-ET in Chinese population : a Meta-analysisPLOS ONE

Dear Dr. Wang,

Thank you for submitting your manuscript to PLOS ONE. After careful consideration, we feel that it has merit but does not fully meet PLOS ONE’s publication criteria as it currently stands. Therefore, we invite you to submit a revised version of the manuscript that addresses the points raised during the review process.

We look forward to receiving your revised manuscript.

Kind regards,

Guglielmo Stabile

Academic Editor

PLOS ONE

3. PLOS requires an ORCID iD for the corresponding author in Editorial Manager on papers submitted after December 6th, 2016. Please ensure that you have an ORCID iD and that it is validated in Editorial Manager. To do this, go to ‘Update my Information’ (in the upper left-hand corner of the main menu), and click on the Fetch/Validate link next to the ORCID field. This will take you to the ORCID site and allow you to create a new iD or authenticate a pre-existing iD in Editorial Manager. Please see the following video for instructions on linking an ORCID iD to your Editorial Manager account: https://www.youtube.com/watch?v=_xcclfuvtxQ.

6. We notice that your supplementary figure is uploaded with the file type 'Figure'. Please amend the file type to 'Supporting Information'. Please ensure that each Supporting Information file has a legend listed in the manuscript after the references list.

Additional Editor Comments:

Please answer reviewer 1's questions correctly. Some aspect of the discussion should be clarified.

Reviewers' comments:

Reviewer's Responses to Questions

**Comments to the Author**

1. Is the manuscript technically sound, and do the data support the conclusions?

Reviewer #1: Partly

Reviewer #2: Yes

2. Has the statistical analysis been performed appropriately and rigorously? 

Reviewer #1: I Don't Know

Reviewer #2: Yes

3. Have the authors made all data underlying the findings in their manuscript fully available?

Reviewer #1: Yes

Reviewer #2: Yes

4. Is the manuscript presented in an intelligible fashion and written in standard English?

Reviewer #1: No

Reviewer #2: Yes

5. Review Comments to the Author

Reviewer #1: Previous ectopic pregnancy, previous pelvic inflammatory disease, pelvic surgery, anf of course the increase in assisted reproductive techniques are responsibles of the increased incidence of ectopic pregnancies (Tubal and non tubal ectopic pregnancies ex. cesarean scar pregnancy; interstitial, cervical pregnancy, ovarian prengnancy.. ). It is important to clarify the difference between the types of ectopic pregnancies, because they are characterized by extremely different rates or morbidity and mortality ( for example interstitial pregnancy has a mortality rate around 2–2.5%, which is seven times the average for all ectopic pregnancies).

Timely diagnosis is the key for successful and conservative management of ectopic pregnancy.

There is no consensus about the best approach to adopt mainly due to a lack of evidence about the best treatment modality after a comparison in large series of clinical cases or randomized studies. You could explain the different strategies of conservative treatments for the different type of ectopic pregnancies, according to the “objctives” of the study “to provide reference for targeted prevention and treatment”. I think you have to implement this aspect in your discussion.

Section Inclusion /exclusion criteria: Line 58 “same risk factor is greater than or equal to 2 articles. ( 9 )” . You have to put “;” .

Figure 1. You have to correct capital letters

Tables 1. Maybe you can change the format because it is not so clear.

Line 144 “egger 's test “ capital letter.

Lines 162-165 “insufficient ovarian reserve and PCOS are risk factors for ectopic pregnancy after IVF-ET, which may be related to the poor ovarian hormone environment in patients with ovarian dysfunction, and the endocrine support of the ovary to the endometrium is difficult to maintain the best state, resulting in poor endometrial receptivity” Please, explain better.

Reviewer #2: Thank you to allow to me to review this interesting manuscript. Even if this review has some limitations (probably the biggest are the inclusion only of studies performed on the Chinese population and the limited number of the studies in literature), it is well written, with a good Discussion on all the risk factors that emerged from the analysis. The conclusions of this review could be very useful to identify an high risk group among women (in chinese Population) that undergoing IVF.

6. PLOS authors have the option to publish the peer review history of their article (what does this mean?). If published, this will include your full peer review and any attached files.

Reviewer #1: No

Reviewer #2: No

---

## [Author Response · Author response to Decision Letter 0]

18 Nov 2023

Response to editor

1.Please ensure that your manuscript meets PLOS ONE's style requirements, including those for file naming.

Response: The format has been revised according to your requirements. 

2.Please update your submission to use the PLOS LaTeX template.

Response: We have made modifications to the article using the PLOS LaTeX template.

3.PLOS requires an ORCID iD for the corresponding author in Editorial Manager on papers submitted after December 6th, 2016. Please ensure that you have an ORCID iD and that it is validated in Editorial Manager. 

Response: We have applied for an ORCID ID and verified it in the editor manager. Our 16-digit ORCID identifier is 0009-0009-3585-6339.

4.Please amend either the abstract on the online submission form (via Edit Submission) or the abstract in the manuscript so that they are identical.

Response: We have made modifications.

5.Please include captions for your Supporting Information files at the end of your manuscript, and update any in-text citations to match accordingly. 

Response: We have already done so.

6. We notice that your supplementary figure is uploaded with the file type 'Figure'. Please amend the file type to 'Supporting Information'. Please ensure that each Supporting Information file has a legend listed in the manuscript after the references list.

Response: We have made modifications according to your requirements.

Response to reviewers 

1.Is the manuscript technically sound, and do the data support the conclusions? 

The manuscript must describe a technically sound piece of scientific research with data that 

supports the conclusions. Experiments must have been conducted rigorously, with appropriate 

controls, replication, and sample sizes. The conclusions must be drawn appropriately based on 

the data presented. 

Reviewer #1: Yes 

Reviewer #2: Partly 

Response: The statistical results in this meta-analysis support this conclusion.

2.Has the statistical analysis been performed appropriately and rigorously? 

Reviewer #1: Yes 

Reviewer #2: I Don't Know 

Response: The statistical methods of this meta-analysis are all operated in strict accordance 

with the guidelines for meta-analysis.

3.Have the authors made all data underlying the findings in their manuscript fully available? 

The PLOS Data policy requires authors to make all data underlying the findings described in their 

manuscript fully available without restriction, with rare exception (please refer to the Data 

Availability Statement in the manuscript PDF file). The data should be provided as part of themanuscript or its supporting information, or deposited to a public repository. For example, in 

addition to summary statistics, the data points behind means, medians and variance measures 

should be available. If there are restrictions on publicly sharing data—e.g. participant privacy or 

use of data from a third party—those must be specified. 

Reviewer #1: Yes 

Reviewer #2: Yes 

Response: We have uploaded my study’s minimal underlying data set as Supporting 

Information files.

4.Is the manuscript presented in an intelligible fashion and written in standard English?

Reviewer #1: No

Reviewer #2: Yes

Response: According to your request, I have polished my language to a certain extent. 

5.Review Comments to the Author

Reviewer #1: Previous ectopic pregnancy, previous pelvic inflammatory disease, pelvic surgery, anf of course the increase in assisted reproductive techniques are responsibles of the increased incidence of ectopic pregnancies (Tubal and non tubal ectopic pregnancies ex. cesarean scar pregnancy; interstitial, cervical pregnancy, ovarian prengnancy.. ). It is important to clarify the difference between the types of ectopic pregnancies, because they are characterized by extremely different rates or morbidity and mortality ( for example interstitial pregnancy has a mortality rate around 2–2.5%, which is seven times the average for all ectopic pregnancies).

Timely diagnosis is the key for successful and conservative management of ectopic pregnancy.

There is no consensus about the best approach to adopt mainly due to a lack of evidence about the best treatment modality after a comparison in large series of clinical cases or randomized studies. You could explain the different strategies of conservative treatments for the different type of ectopic pregnancies, according to the “objctives” of the study “to provide reference for targeted prevention and treatment”. I think you have to implement this aspect in your discussion.

Section Inclusion /exclusion criteria: Line 58 “same risk factor is greater than or equal to 2 articles. ( 9 )” . You have to put “;” .

Figure 1. You have to correct capital letters

Tables 1. Maybe you can change the format because it is not so clear.

Line 144 “egger 's test “ capital letter.

Lines 162-165 “insufficient ovarian reserve and PCOS are risk factors for ectopic pregnancy after IVF-ET, which may be related to the poor ovarian hormone environment in patients with ovarian dysfunction, and the endocrine support of the ovary to the endometrium is difficult to maintain the best state, resulting in poor endometrial receptivity” Please, explain better.

Response: According to your requests, we clarified the differences between ectopic pregnancies in the discussion and also added differences in conservative treatment strategies for different types of ectopic pregnancies. We have changed lines 162-165 to “Decreased ovarian reserve and PCOS are also risk factors for EP after IVF-ET. This connection may be related to ovarian endocrine dysfunction and dysregulation of estrogen and progesterone levels in these patients, resulting in poor endometrial receptivity formed by the combined action of estrogen and progesterone”. We have changed the capital letters in Figure 1 and Line 144, and adjusted the format of Table 1.

Some of our major modifications: 

1.We have polished our language to a certain extent.

2. At the request of the reviewer, some contents were added to the discussion, and ten 

references were added. 

3.We have adjusted the order of authors

---

## [Decision Letter · Decision Letter 1]

15 Dec 2023

Risk factors of ectopic pregnancy after in vitro fertilization-embryo transfer in Chinese population: A meta-analysis

PONE-D-23-29164R1

Dear Dr. Yanbo Wang

We’re pleased to inform you that your manuscript has been judged scientifically suitable for publication and will be formally accepted for publication once it meets all outstanding technical requirements.

Kind regards,

Guglielmo Stabile

Academic Editor

PLOS ONE

Additional Editor Comments (optional):

The manuscript has been improved. In my opinion is now suitable for publication

Reviewers' comments:

Reviewer's Responses to Questions

**Comments to the Author**

1. If the authors have adequately addressed your comments raised in a previous round of review and you feel that this manuscript is now acceptable for publication, you may indicate that here to bypass the “Comments to the Author” section, enter your conflict of interest statement in the “Confidential to Editor” section, and submit your "Accept" recommendation.

Reviewer #1: (No Response)

2. Is the manuscript technically sound, and do the data support the conclusions?

Reviewer #1: Yes

3. Has the statistical analysis been performed appropriately and rigorously? 

Reviewer #1: N/A

4. Have the authors made all data underlying the findings in their manuscript fully available?

Reviewer #1: Yes

5. Is the manuscript presented in an intelligible fashion and written in standard English?

Reviewer #1: Yes

6. Review Comments to the Author

Reviewer #1: (No Response)

7. PLOS authors have the option to publish the peer review history of their article (what does this mean?). If published, this will include your full peer review and any attached files.

Reviewer #1: **Yes: **Giulia Zinicola

---

## [Editor Report · Acceptance letter]

21 Dec 2023

PONE-D-23-29164R1 

PLOS ONE

Dear Dr. Wang, 

I'm pleased to inform you that your manuscript has been deemed suitable for publication in PLOS ONE. Congratulations! Your manuscript is now being handed over to our production team.

Kind regards, 

on behalf of

Dr. Guglielmo Stabile 

Academic Editor

PLOS ONE